# The utility of serology for elimination surveillance of trachoma

Amy Pinsent[1,2], Anthony W. Solomon[3,4], Robin L. Bailey[4], Rhiannon Bid[4], Anaseini Cama[5,6], Deborah Dean[7], Brook Goodhew[8], Sarah E. Gwyn[8], Kelvin R. Jack[9], Ram Prasad Kandel[10], Mike Kama[11], Patrick Massae[12], Colin Macleod[4,13], David C.W. Mabey[4], Stephanie Migchelsen[4], Andreas Müller[14], Frank Sandi[12,15], Oliver Sokana[9], Raebwebwe Taoaba[16], Rabebe Tekeraoi[16], Diana L. Martin[8] & Michael.T. White[17]

Robust surveillance methods are needed for trachoma control and recrudescence monitoring, but existing methods have limitations. Here, we analyse data from nine trachoma-endemic populations and provide operational thresholds for interpretation of serological data in low-transmission and post-elimination settings. Analyses with sero-catalytic and antibody acquisition models provide insights into transmission history within each population. To accurately estimate sero-conversion rates (SCR) for trachoma in populations with high-seroprevalence in adults, the model accounts for secondary exposure to *Chlamydia trachomatis* due to urogenital infection. We estimate the population half-life of sero-reversion for anti-Pgp3 antibodies to be 26 (95% credible interval (CrI): 21–34) years. We show SCRs below 0.015 (95% confidence interval (CI): 0.0–0.049) per year correspond to a prevalence of trachomatous inflammation—follicular below 5%, the current threshold for elimination of active trachoma as a public health problem. As global trachoma prevalence declines, we may need cross-sectional serological survey data to inform programmatic decisions.

[1] Department of Public Health and Preventative Medicine, Monash University, Melbourne, VIC 3004, Australia. [2] Department of Infectious Disease Epidemiology, London School of Hygiene & Tropical Medicine, London WC1E 7HT, UK. [3] Department of Control of Neglected Tropical Diseases, World Health Organization, 1211 Geneva 27, Switzerland. [4] Clinical Research Department, London School of Hygiene & Tropical Medicine, London WC1E 7HT, UK. [5] International Agency for the Prevention of Blindness, Western Pacific Region, Suva, Fiji. [6] The Fred Hollows Foundation, Level 2, 61 Dunning Ave, Rosebury, NSW 2018, Australia. [7] UCSF Benioff Children's Hospital Oakland Research Institute, 5700 Martin Luther King Jr Way, Oakland, CA 94609, USA. [8] Division of Parasitic Diseases and Malaria, Centers for Disease Control and Prevention, 1600 Clifton Road, Atlanta, GA 30333, USA. [9] Eyecare Department, Ministry of Health, Honiara, Solomon Islands. [10] Lumini Eye Hospital, Bhairahawa, Nepal. [11] Department of Communicable Diseases, Ministry of Health, Suva, Fiji. [12] Department of Ophthalmology, Kilimanjaro Christian Medical Centre, Moshi, Tanzania. [13] Sightsavers, 35 Perrymount Road, Haywards Heath RH16 6NG, UK. [14] Centre for Eye Research Australia, Level 7/32 Gisborne St, East Melbourne, VIC 3002, Australia. [15] The University of Dodoma, Dodoma, Tanzania. [16] Eye Department, Ministry of Health and Medical Services, South Tarawa, Kiribati. [17] Malaria: Parasites & Hosts, Department of Parasites and Insect Vectors, Institut Pasteur, 25-28 Rue du Dr Roux, 75015 Paris, France. Correspondence and requests for materials should be addressed to A.P. (email: amy.pinsent@lshtm.ac.uk)

Trachoma is a neglected tropical disease (NTD) caused by repeated infection with the bacterial pathogen *Chlamydia trachomatis* (*Ct*)[1] and is targeted for elimination as a public health problem by 2020. In 2016, an estimated 190.2 million people were at risk of trachomatous blindness in 41 countries[2]. There has been substantial progress towards achieving elimination of trachoma as a public health problem[2], with many countries accelerating towards the active trachoma target of a prevalence of trachomatous inflammation—follicular (TF) <5% in all previously-endemic districts.

Following elimination of trachoma as a public health problem, robust surveillance for disease recrudescence will be needed. Currently, programmatic decisions for trachoma control—specifically, for the number of rounds of azithromycin mass drug administration (MDA) rounds required before re-estimation of prevalence, and for MDA discontinuation—are made based on TF prevalence in children aged 1–9 years. Once a TF prevalence of <5% has been achieved, a pre-validation surveillance survey is required to assess whether re-emergence is occurring or not; if TF prevalence is found to be ≥5%, re-initiation of MDA would be indicated. At low levels of TF prevalence, there is no clear association between the prevalence of TF and ocular *Ct* infection[3], making the interpretation of data difficult, and suggesting that different diagnostics may be required at particular phases of programmatic monitoring and evaluation. Furthermore when active trachoma prevalence falls, it becomes difficult to adequately train graders and prove they are proficient at identifying TF[4]. There is therefore a clear need for surveillance methods that accurately monitor low levels of transmission, and serology is potentially one such method.

Identification of optimal approaches for surveillance following the cessation of interventions remains an on-going challenge for numerous infectious diseases[5,6]. Serological assays measuring antibody responses resulting from a single or cumulative exposure to a pathogen have been used to measure and assess changes in transmission intensity[7], including examining the impact of interventions on transmission[8]. Such testing can potentially be integrated into existing surveillance mechanisms. Modelling sero-epidemiological data to understand patterns of transmission is well-established in influenza and malaria epidemiology[6,9], and is increasingly performed for other infectious diseases, including onchocerciasis[10], Chagas disease[11], and lymphatic filariasis[12]. Given the insights provided through the analysis of serological data for other pathogens, serology has been suggested as a complementary or alternative surveillance tool to the use of clinical signs in trachoma programmes. Antibodies against two *Ct*-derived antigens (Pgp3 and CT694) are detectable in a very high proportion of people with ocular *Ct* infection[7]. However, Pgp3 and CT694 are shared by *Ct* serovars associated with urogenital infection, complicating our understanding of patterns of ocular *Ct* transmission[13,14]. If an accurate understanding of ocular *Ct* transmission is to be inferred from analysis of population-level serological data, a second potential source of exposure to *Ct* antigens via urogenital infection must be considered.

To evaluate the utility of serology as a tool for early detection of recrudescence, evidence must be analysed from a range of epidemiological settings[15]. Particular requirements for results from sero-surveillance to be informative include an adequate understanding of the population-level sero-reversion rate (SRR) and antibody dynamics, so changes in transmission can be monitored post-validation.

The collection and analysis of *Ct* serology data is an ongoing and active area of trachoma research. A previous modelling analysis of serological data from Rombo, Tanzania suggested that a step-wise drop in transmission occurred ~15 years prior to the survey date[16]. Equally, a study analysing data from The Gambia also suggested that a step-wise drop in transmission occurred

19–23 years prior to sampling[14]. However, in analyses to date, no consideration has been given to age-dependent exposure to urogenital *Ct* when estimating the sero-conversion rate (SCR). Furthermore, previous studies have either assumed that sero-reversion following conversion does not occur at all[16] or that it takes on average at least 65 years[14] and estimates of this parameter have been limited to data from one cross-sectional survey. Therefore, more research is required in order to estimate the SRR from multiple cross-sectional surveys, and to estimate the SCR whilst accounting for the potential exposure to urogenital *Ct* in settings where this may be a problem.

In this study, we explore age-specific variation in antibody responses to *Ct*. We demonstrate how sero-catalytic and antibody acquisition models provide insight into current and historical patterns of trachoma transmission. We quantify the relationship between the estimated *Ct* sero-prevalence and TF prevalence to help guide the framing of operational thresholds for sero-surveillance data.

## Results

**Overview of the analysis.** We performed analysis on 9 data sets from 6 different geographic regions and analysed the data using two different statistical models: sero-catalytic models and antibody acquisition models. We evaluated age-dependent changes in anti-Pgp3 and anti-CT694 antibody prevalence to infer historical patterns of transmission within each setting. For each of the different model types (sero-prevalence and antibody acquisition) three distinct transmission scenarios were considered: scenario 1 assumed a constant rate of transmission; scenario 2 assumed a sharp drop in transmission $t_c$ years ago; and scenario 3 assumed a linear decline in transmission[8,17]. We fitted up to 10 different transmission scenarios to each dataset. To understand the relationship between different measures of transmission intensity (SCR and TF prevalence) we fitted a linear model to the relationship between the SCR for trachoma ($\lambda_T$) (for Pgp3) and TF prevalence for each study site. Full details on the methodology are provided in the Methods section and Supplementary Methods.

**Correlation between antibody responses against two Ct antigens and age.** We observed a strong positive correlation between antibody responses against the two antigens in Nepal, both pre- and post-MDA, and in Rombo (correlation coefficients 0.80, 0.85, 0.83 for the three data sets, respectively). For all study sites there was a strong positive correlation between age and antibody response (measured in MFI-BG or OD), with older individuals having higher antibody responses than younger individuals (Supplementary Figure 1).

**Sero-catalytic modelling.** In Fig. 1 we present the sero-prevalence, TF prevalence and the fit of the most suitable sero-catalytic model scenario for each dataset. Parameter estimates for the best-performing model for each of the nine data sets are presented in Table 1. Full sets of parameter estimates for every model fitted are presented in Supplementary Tables 3–13 and Supplementary Figures 3-13.

Sero-catalytic models were fitted to the anti-Pgp3 and anti-CT694 antibody data from the pre- and post-MDA cross-sections from Nepal simultaneously. The data were best described by scenario $2_{v1}$: a step-wise decrease in transmission (Fig. 1a, Supplementary Table 3). Antibody responses to both antigens had similar estimates of $\lambda_T$: 0.143 (95% credible interval (CrI): 0.107–0.215) per year and 0.142 (95% CrI: 0.103–0.207) per year for Pgp3 and CT694, respectively (Table 2). Antibody responses against Pgp3 and CT694 had differing longevity, with the half-life for sero-reversion estimated as 26 (95% CrI: 21–34) years, and 40 (95% CrI: 33–53) years for Pgp3 and CT694, respectively. After

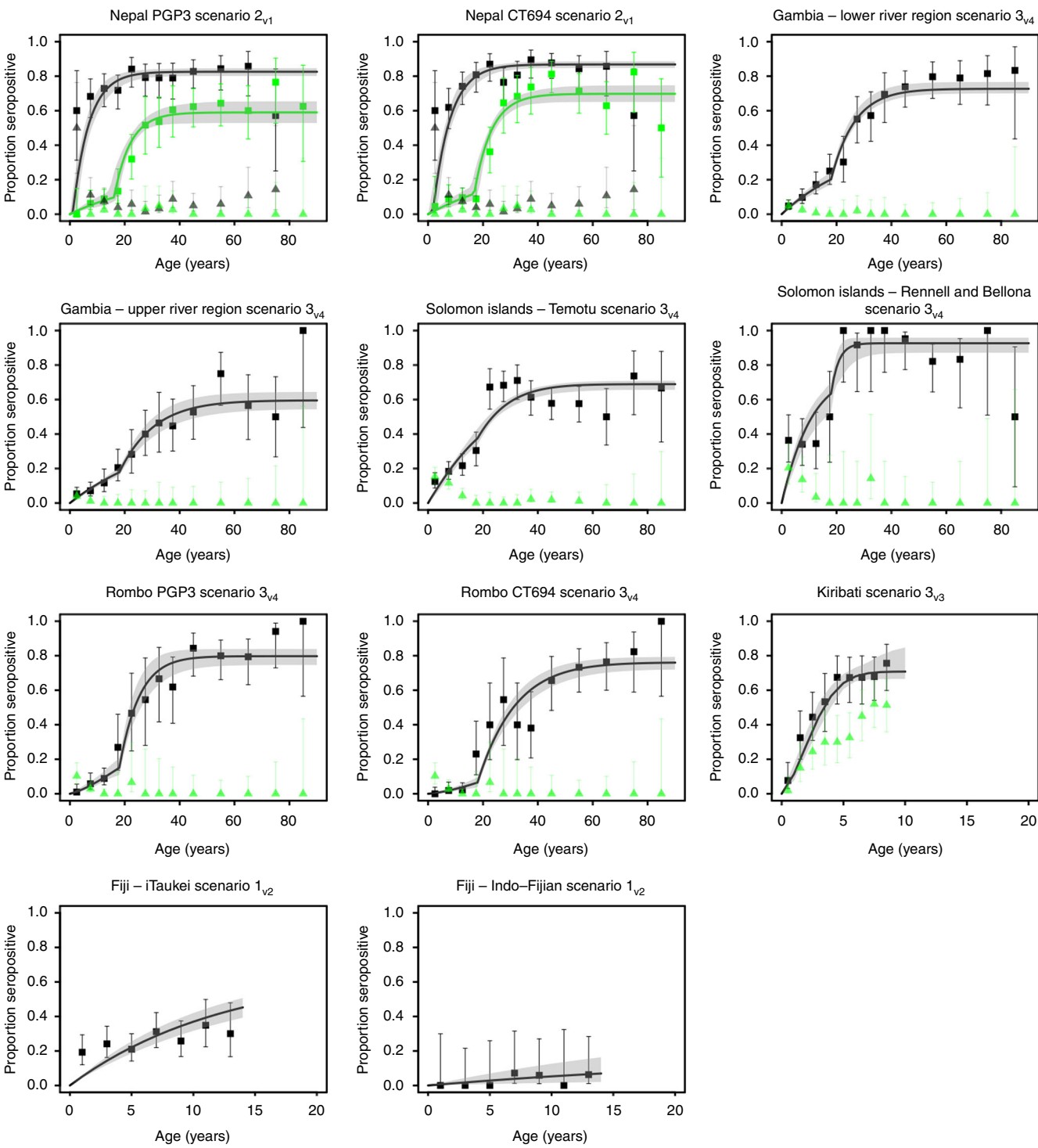

**Fig. 1** Fits of the best-performing sero-catalytic models to age-specific sero-prevalence data. The titles within each panel indicate the study site, antigen-specific antibody responses measured and the best fitting transmission scenario for that dataset. Black squares indicate the proportion sero-positive in each age-group and green triangles indicate the age-group specific TF prevalence. Black and green data points on the Nepal plots indicate pre and post-MDA, respectively. Error bars on the squares and triangles indicate the 95% binomial confidence intervals. Solid black lines running through the sero-prevalence data were generated with the median parameter estimates from each model fit. The shaded grey region represents the 95% credible intervals of the model predictions. Uncertainty was generated by drawing 500 independent samples from the posterior distribution

MDA, transmission was reduced by 5 and 10% of pre-intervention levels, with this difference being consistent when using antibody responses to either antigen. The estimated time of change in transmission ($t_c$) was 13–17 years prior to the second cross-section using antibody responses to Pgp3, and 14–18 years prior to the second cross-section using antibody responses to CT694.

Data from Lower and Upper River regions (LRR and URR) of The Gambia, from Rennell & Bellona and Temotu provinces in the Solomon Islands, and from Rombo (both antigens) were all fitted separately. For all populations, scenario $3_{v4}$ was most appropriate (a linear reduction in transmission) with fixed $\rho$, also accounting for exposure to urogenital $Ct$ at sexual debut (Fig. 1c–h). This pattern

**Table 1 Definitions of each transmission scenario and the parameters estimated from the data**

| Model name | Transmission assumption | $\lambda_T$ | $\lambda_{UG}$ | $\rho$ | $t_c$ | $\gamma$ |
|---|---|---|---|---|---|---|
| Scenario $1_{v1}$ | Constant | Yes | No | Yes | No | No |
| Scenario $1_{v2}$ | Constant | Yes | No | No | No | No |
| Scenario $2_{v1}$ | Fixed change point | Yes | No | Yes | Yes | Yes |
| Scenario $2_{v2}$ | Fixed change point | Yes | No | No | Yes | Yes |
| Scenario $2_{v3}$ | Fixed change point | Yes | Yes | Yes | Yes | Yes |
| Scenario $2_{v4}$ | Fixed change point | Yes | Yes | No | Yes | Yes |
| Scenario $3_{v1}$ | Linear decline | Yes | No | Yes | No | Yes |
| Scenario $3_{v2}$ | Linear decline | Yes | Yes | Yes | No | Yes |
| Scenario $3_{v3}$ | Linear decline | Yes | No | No | No | Yes |
| Scenario $3_{v4}$ | Linear decline | Yes | Yes | No | No | Yes |

For each scenario we indicate with a yes or a no as to whether or not a given parameter was estimated from the data for that scenario. The parameters listed are defined as follows: $\lambda_T$ (Rate of sero-conversion due to exposure to trachoma), $\lambda_{UG}$ (Rate of sero-conversion due to exposure to urogenital infection), $\rho$ (Rate of sero-reversion), $t_c$ (Fixed time point at which transmission intensity changed), $\gamma$ (Proportional decline in transmission at $t_c$ or over time). Note that the interpretation of $\gamma$ for scenario 2 and 3 are different. For scenario 2, it is the ratio between average transmission rates of two time intervals. For scenario 3, it is the ratio between two end points of the whole study period

**Table 2 Estimated parameters for the best fitting sero-catalytic models to each of the 9 data sets**

| Study site | Model | $\lambda_T$ | $\gamma$ | $p$ | $t_c$ | $\lambda_{UG}$ | DIC |
|---|---|---|---|---|---|---|---|
| Nepal (Pgp3) | Scenario $2_{v1}$ | 0.143 (0.107–0.215) | 0.053 (0.031–0.084) | 0.026 (0.020–0.032) | 16.06 (13.57–17.69) | — | 1325.11 |
| Nepal (CT694) | Scenario $2_{v1}$ | 0.142 (0.103–0.207) | 0.062 (0.037–0.097) | 0.017 (0.013–0.021) | 16.84 (14.93–18.65) | — | 1252.95 |
| Gambia LRR | Scenario $3_{v4}$ | 0.021 (0.013–0.03) | 0.677 (0.268–0.984) | — | — | 0.067 (0.049–0.090) | 866.64 |
| Gambia URR | Scenario $3_{v4}$ | 0.023 (0.010–0.184) | 0.591 (0.112–0.893) | — | — | 0.063 (0.015–0.595) | 678.64 |
| Rombo (Pgp3) | Scenario $3_{v4}$ | 0.022 (0.009–0.041) | 0.177 (0.019–0.881) | — | — | 0.092 (0.061–0.127) | 379.74 |
| Rombo (CT694) | Scenario $3_{v4}$ | 0.008 (0.004–0.016) | 0.172 (0.002–0.567) | — | — | 0.048 (0.031–0.062) | 326.74 |
| Temotu | Scenario $3_{v4}$ | 0.045 (0.028–0.075) | 0.585 (0.218–0.986) | — | — | 0.021 (0.001–0.041) | 1440.54 |
| Rennell & Bellona | Scenario $3_{v4}$ | 0.092 (0.061–0.170) | 0.746 (0.319–0.995) | — | — | 0.255 (0.085–0.499) | 247.34 |
| Kiribati | Scenario $3_{v3}$ | 1.080 (0.345–1.737) | 0.063 (0.034–0.204) | — | — | — | 453.45 |
| iTaukei | Scenario $1_{v2}$ | 0.053 (0.044–0.063) | — | — | — | — | 554.99 |
| Indo-Fijian | Scenario $1_{v2}$ | 0.006 (0.001–0.014) | — | — | — | — | 23.30 |

We present the median posterior estimates, the 2.5% and 97.5% credible intervals (CrI) for each parameter for each model and the Deviance information criteria (DIC) for each model (note that DIC values should not be compared between different model fits to different data sets). $\lambda_T$ - rate of sero-conversion due to exposure to trachoma, $\lambda_{UG}$ - rate of sero-conversion due to exposure to urogenital infection, $\rho$ - rate of sero-reversion, $t_c$ - fixed time point at which transmission intensity changed, $\gamma$ - proportional decline in transmission at $t_c$ or over time. Lower River Region (LRR), Upper River Region (URR)

corresponds to declining trachoma transmission over time (akin to a secular decline) without a sharp reduction. Sero-prevalence curves observed in Rombo and The Gambia were very similar to one another, suggesting the locations are epidemiologically similar. In both settings scenario $2_{v1}$ was either statistically comparable or better performing based on the Deviance Information Criterion (DIC). Although model estimates are consistent with declining transmission in these regions, the exact magnitude of this reduction was not identifiable because of the simultaneous age-dependent exposure to urogenital infection (Table 2).

Data sets from Kiribati and Fiji included only individuals <16 years old. In Kiribati, we saw increasing sero-prevalence by age (Fig. 1i). No increase in sero-positivity with age was observed in Fiji (Fig. 1j, k). Scenario $3_{v3}$ was most suitable for Kiribati (Fig. 1i, Supplementary Table 11). $\lambda_T$ was 0.32 (95% CrI: 0.17–0.52) per year and $\gamma$ was 0.063 (95% CrI: 0.034–0.204) (Table 2). In Fiji all scenarios provided comparably poor fits to the data and there was no epidemiological evidence to support the scenario that had the highest statistical support (Scenario $2_{v2}$ for both ethnic populations, Supplementary Table 12 & 13). Therefore scenario $1_{v2}$ (assuming a constant level of transmission with a fixed $\rho$) was selected as the most parsimonious model (Table 2 and Supplementary Table 12 & 13). Low rates of $\lambda_T$ were estimated: 0.053 (95% CrI: 0.044–0.063) per year and 0.006 (95% CrI: 0.001–0.014) per year for the iTaukei and Indo-Fijian populations respectively (Table 2).

**Antibody acquisition modelling**. Fitting the antibody acquisition (AA) model to the data from Nepal, greater statistical support was provided for scenario $2_{v1}$ (Fig. 2, Supplementary Tables 14 & 15). The estimated rate of antibody acquisition ($\alpha_T$) was 0.362 (95% CrI: 0.327–0.403) per year for Pgp3, and 0.373 (95% CrI: 0.329–0.437) per year for CT694 (Supplementary Tables 13 & 14). The estimated antibody level half-life (accounting for both the decay of circulating IgG and the generation of new IgG) was 7.3 (95% CrI: 6.5–8.2) years and 5.5 (95% CrI: 4.7–6.3) years for Pgp3 and CT694, respectively, a shorter duration than the half-life of sero-positivity. Model fits to data sets with a single cross-section are provided in the Supplementary Tables 14–24 and Supplementary Figures 14–22, discussion of these results are provided in the Supplementary discussion.

**Relationship between $\lambda_T$ and TF prevalence**. The association between TF prevalence (in 1–9 year olds) and the SCR was captured using a linear model (Fig. 3a). The relationship provides a bridge between estimates of transmission intensity based on either TF prevalence data or serological data. The predicted $\lambda_T$ when TF prevalence was <5% was 0.015 (95% CI: 0.0–0.051) per year (Fig. 3a). The expected proportion of sero-positive individuals at a TF prevalence of <5% was 6.2% (95% CI: 0.0–19.9%) (Fig. 3b).

**Assessing sub-critical transmission in serological data**. Figure 4a presents a simulated dataset from a population aged 1–60 years, 10 years after elimination of trachoma as a public health problem. Older individuals have higher antibody positivity, as a result of either long-lived response to ocular $Ct$ or exposure to urogenital $Ct$ after trachoma's elimination as a public health problem, with much lower or no sero-positivity in children. Assessing individuals aged 1–9 years born after trachoma has been eliminated

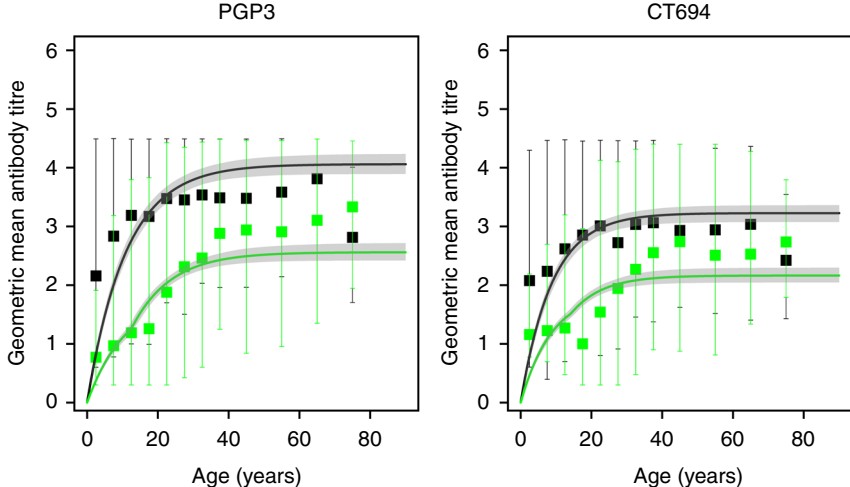

**Fig. 2** Fits of the best-performing antibody acquisition model for data from Nepal. Black points indicate the pre-MDA data and green indicate the post-MDA data. Error bars on the squares and triangles indicate the 95% binomial confidence intervals. Solid black lines running through the sero-prevalence data were generated with the median parameter estimates from each model fit. The shaded grey region represents 95% credible intervals of the model predictions. Uncertainty was generated by drawing 500 independent samples from the posterior distribution

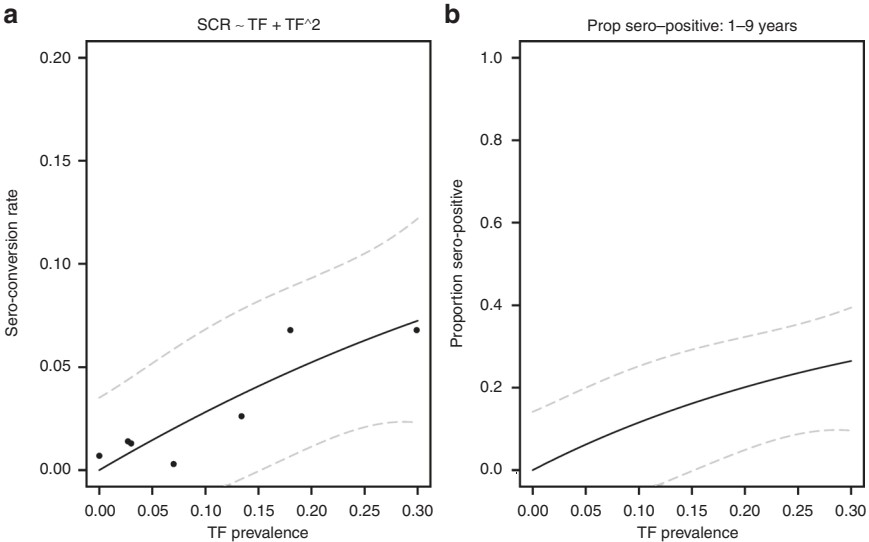

**Fig. 3** The estimated relationship between the sero-conversion rate (SCR) and TF prevalence and the predicted proportion of people sero-positive. **a** Black dots indicate the median estimated SCR for each dataset and the TF prevalence from each of the 9 study sites. The solid black line is the mean predicted relationship between the SCR and TF prevalence, obtained by fitting a linear model to the data. The 95% confidence intervals about the mean relationship are indicated as grey dashed lines. **b** The predicted mean proportion of people sero-positive for a given level of TF prevalence is shown with a solid black line, the 95% confidence intervals about this mean are indicated with dashed grey lines. For the elimination as a public health problem threshold of TF <5%, we would expect 6.2% (95% CI: 0.0–19.9%) to test sero-positive

as a public health problem (Fig. 4b), one of three scenarios may be observed. The first scenario shows no young individuals testing sero-positive (black line). The second scenario shows a slight increase in sero-positivity with age (pink line) due to on-going low-level *Ct* transmission despite TF being below the 5% threshold. The third scenario shows constant, non-zero sero-prevalence across the 1–9 year old age bracket (blue line), reflecting either low-level but not constant transmission or cross-reactivity resulting in a non-specific measurement of antibody response.

**Sampling for elimination as a public health problem.** Figure 4c presents the number of samples in 1–9 year olds required to provide statistically robust evidence that sero-prevalence was below a given threshold, assuming a type 1 error of 5%. In a

population with no *Ct*-sero-positive individuals, at least 368 sero-negative samples are needed to provide statistically significant evidence that sero-prevalence was <1%, assuming random sampling of the underlying population. Demonstrating sero-prevalence below the lower threshold of 0.1% would require an impractical 3766 negative samples. In contrast, demonstrating evidence of sero-prevalence <7% would require only 51 sero-negative samples. However, these estimates of sample numbers are contingent on all samples testing sero-negative. For example, although obtaining 0/51 sero-positive samples provides statistical support for sero-prevalence <7%, just one sero-positive sample would give an estimated sero-prevalence of 1/51 = 2%. This is <7%, but not significantly lower than 7%. Increasing sample numbers allows for some samples to test positive, with estimated

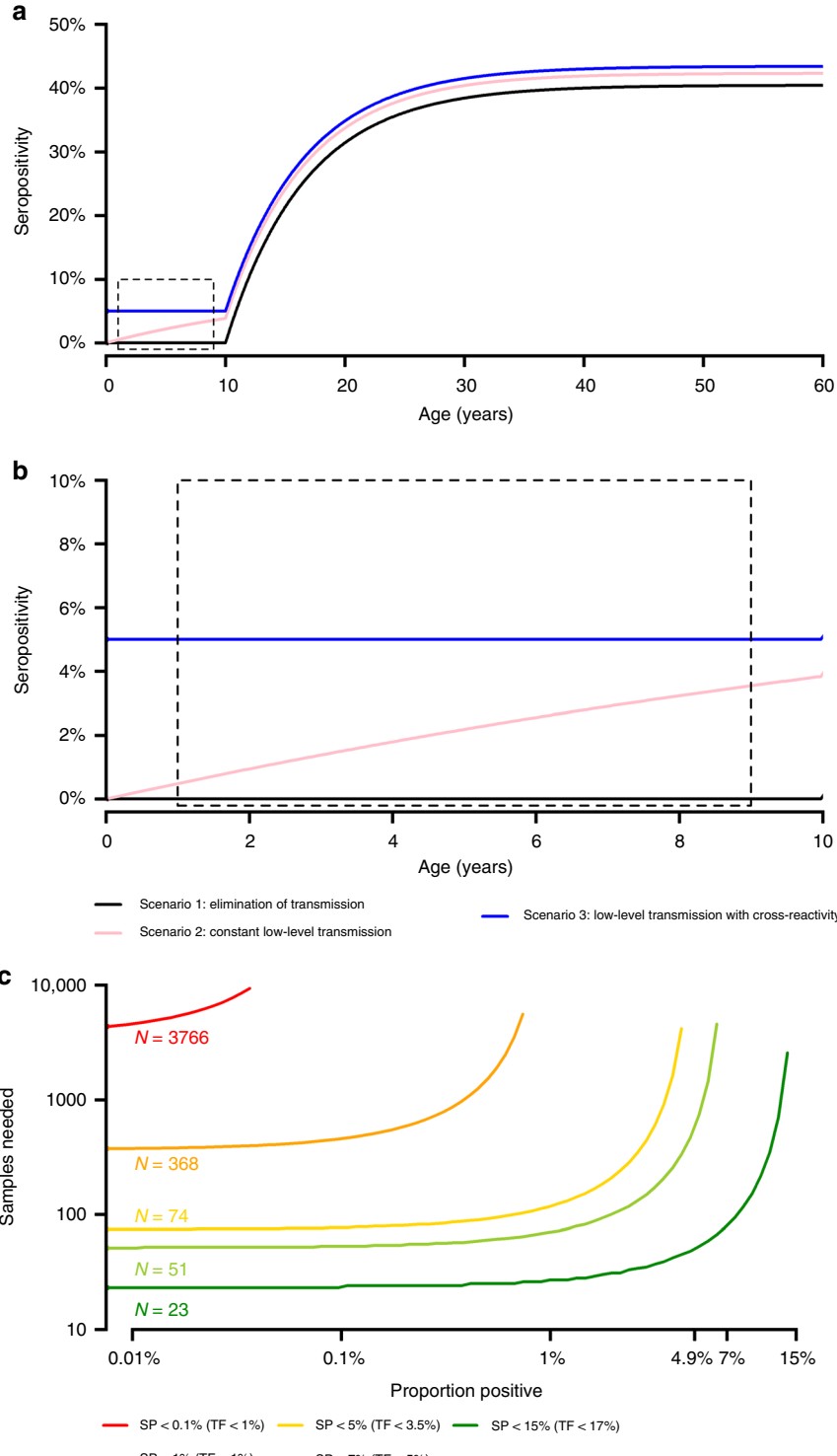

**Fig. 4** Modelled age-specific sero-prevalence curves obtained from a community post-elimination. **a** Scenarios for different average age-specific sero-prevalence curves post-elimination in individuals aged 1–60 years old. Each coloured line represents possible data that may be collected following elimination. **b** A close up of the data presented in (**a**) of the average age sero-prevalence data in individuals only aged 1–9 years old. Possible scenarios are that on average there is no age-specific variation in sero-positivity by age (blue line), there is a slight but not substantial increase in sero-positivity with age (pink line), or no sero-positive individuals in the community at all reflecting complete elimination (black line). **c** Estimate of the number of samples required from children aged 1–9 years to provide statistical evidence that sero-prevalence is below thresholds of: 0.1%, 1%, 4.9%, 7 and 15%. If the true sero-prevalence = 0% such that all samples test negative, the number of samples required is shown where the curves intersect the y-axis. In the situation where there is some low level of transmission, the number of samples increases substantially. For example, if the true sero-prevalence = 0.5%, then 368 samples are needed to provide evidence of sero-positivity <1%

sero-prevalence still significantly lower than 7%. With 100 samples, we can have up to 1 sero-positive sample for sero-prevalence estimates significantly lower than 7%. For 200 samples, the number of allowable sero-positive samples rises to 6.

## Discussion

Serological surveillance for trachoma is being considered to help programmes undertake post-validation surveillance. Prior analyses of the potential utility of serology have used only single-study-site data, with few studies taking a quantitative modelling approach[14,18]. In all settings evaluated, our modelling suggested transmission of trachoma had declined over time, either through a step-change in transmission, or more linearly. We estimated SRR half-life to be 26 (95% CrI: 21–34) years and 40 (95% CrI: 33–53) years, with an estimated half-life of the antibody response to be 7.3 (95% CrI: 6.5–8.2) years and 5.5 (95% CrI: 4.7–6.3) years, for Pgp3 and CT694, respectively—the first estimates published for these parameters. Our results suggested that SCRs below 0.015 per year correspond to TF <5%, and that the mean sero-prevalence for 1–9 year olds when TF <5% was <7%. Whilst more data are needed to reduce the uncertainty in the relationship between sero-prevalence and TF prevalence, we present an initial operational threshold for interpreting serological data in trachoma programmes.

In most settings, the sero-prevalence of anti-$Ct$ antibodies increased with age. A typical ocular $Ct$ infection commonly lasts only a few weeks or months, which may or may not be sufficient to develop long-lived plasma cells that secrete antibodies for many years, even in the absence of subsequent infection that would provide boosting of the response. Though it is impossible to rule out bacterial persistence in tissues, it is more likely that the increase in sero-prevalence with increasing age is due to repeated infection with ocular $Ct$ leading to the induction of plasma cells[19–21]. Therefore, we believe that the measured response reflects cumulative exposure to the pathogen and immunological memory.

We have presented a novel approach to account for secondary (non-trachoma-related) antigen exposure that arises through infection with urogenital $Ct$, allowing more accurate estimation of the SCR for trachoma. The framework presented for assessing cross-sectional serological samples and its link to measures of TF prevalence has important implications for trachoma surveillance in an era of declining prevalence[22]. Modelling suggested that transmission had declined in all regions. Validation of this could be undertaken by comparing these findings with longitudinal trachoma surveillance data. We would expect to see estimates of declining transmission reflected by reductions in Ct PCR positivity and TF prevalence between the two cross-sections. In populations from which historical data (or multiple cross-sections) were available for this study (Nepal, The Gambia and Rombo) we see clear declines in TF and PCR prevalence (where available) mirroring the serological pattern[14,18,23]. For future studies, it will be important to ensure that long-term TF and PCR data are generated in order to validate sero-surveillance data. In regions aiming to eliminate trachoma as a public health problem, there are key opportunities for serological data to contribute to future decisions on when to stop, or if necessary, when to restart MDA.

If, as intimated above, we need TF data to help validate serology, why not just keep examining children for TF instead? Estimation of the SCR may help overcome some of the limitations of using TF prevalence for post-validation surveillance, without requiring larger sample sizes. As transmission intensity declines within a population, the specificity of TF for ocular $Ct$ infection decreases[3] and it becomes more costly to train graders to identify TF[24]. Moreover, age-specific TF prevalence patterns can be highly variable, whilst age-specific sero-prevalence patterns observed were much more consistent, with the observed variation in sero-prevalence reflecting historical exposure patterns. We therefore contend that SCR in 1–9 year olds can provide a consistent measurement of transmission intensity. Photography (of eyelids, with later grading of photographs for TF) combined with photographer training, has also been suggested as a potential surveillance approach that might be more cost effective than training field graders[25], but evidence is mounting to suggest that drivers other than $Ct$ may cause the TF phenotype[26]. We have modelled post-elimination serological data under various scenarios. More surveys are still needed to generate examples of sero-prevalence profiles from populations after elimination as a public health problem has been achieved, and to assess how sero-prevalence profiles might be modified by differing incidence of urogenital $Ct$. Using our mean estimate of sero-prevalence of 7% when TF <5%, a sample size of at least 51 sero-negative samples would be needed to confirm maintenance of elimination as a public health problem. However, confirming sero-prevalence levels <1% would require substantially larger sample sizes.

There are several limitations to our study. First, for some study sites, it was difficult to make statistically valid distinctions between models that accounted for a secondary (urogenital) exposure to $Ct$ antigens, and those predicting a fixed change point in trachoma transmission intensity. This was exemplified by the Solomon Islands data, where exposure to urogenital infection likely confounds any estimated $t_c$, (reflected in the uncertain $t_c$ estimates). Therefore, when comparing scenarios within each study site, we used our understanding of the epidemiology and history of interventions to select the most appropriate scenario. Second, we used the estimated SRR from Nepal as our fixed SRR estimate for all other study sites, but the SRR may vary between populations. Moreover, a population-level estimate of the SRR may not be appropriate for young children, as they tend to have lower antibody levels and may sero-revert more quickly than adults. We may need multiple cross-sectional data sets from children within the same population to quantify the SRR. We did not account for secondary exposure to urogenital infection in Nepal. Fortunately, urogenital $Ct$ has a low prevalence in the general population there[27], an observation borne out by the pre-MDA serological data, where there was no sharp age-dependent increase in sero-prevalence that would be consistent with $Ct$ exposure following sexual debut. Third, different study sites used different platforms to assess each individual's antibody response, and it was consequently not possible to directly compare the antibody acquisition rates against TF prevalence; this problem does not arise when comparing results of the sero-catalytic models. Estimates of the best-performing scenario differed slightly when comparing the sero-catalytic and antibody acquisition models; however, given the issue of confounding related to estimating $t_c$ and the time at which individuals may be exposed to urogenital infection, this is not unexpected. Fourth, given the very high prevalence of urogenital infection in the Solomon Islands[28] and high sero-prevalence in young children[26] we cannot discount the possibility that these children may have been born to mothers with urogenital infection and exposed to $Ct$ in the birth canal. In Fiji, sero-prevalence did not increase with age, suggesting children there might experience constant low-level $Ct$ exposure balancing the effect of natural sero-reversion; this $Ct$ exposure in children might, for example, be due to para-trachoma. Contemporaneous TF prevalence data were not available for this study site to help further the interpretation of our findings[29]. Fifth, whilst the number of data points evaluated here is substantially larger than any previous study, data are still limited. There was therefore a reasonably high level of uncertainty for the estimated relationship between SCR and TF prevalence, and

caution should be exercised in drawing definitive conclusions. Lastly, we only considered constant age-specific forces of infection, which may not truly reflect long-term transmission patterns. There was not sufficient statistical power to consider more complex temporal trends.

In order to further understand and provide additional verification of the findings of this study, we offer several considerations for future work. First, an increase in the number of sero-surveillance studies conducted within countries that previously had trachoma as a public health problem are needed. This would provide greater insight into Pgp3 and CT694 sero-prevalence at sub-critical levels of trachoma transmission. Second, increasing the number of samples collected from individuals outside of the 1–9 years of age indicator group, with multiple cross-sections conducted several years apart from one another, would help to ensure the estimated SRR is correct. Third, in order to test the predictions currently made by our models, we would seek to increase the quantity of serological data available from regions very close to the TF <5% target. From these data we could explore how much variation there is in SCRs in populations close to the threshold for TF's elimination as a public health problem.

It remains challenging to correlate serological findings and (1) integrate them with results that have been generated from existing, more-established data on disease, and (2) use them to inform re-evaluation of current guidelines[30]. However, here we have demonstrated the potential utility of antibody-based surveillance to help monitor low-levels of ocular $Ct$ infection transmission. We have initiated an evidence base for the use of sero-surveillance by programmes in low-transmission and post-elimination settings (where TF <5%), and have provided an operational threshold for sero-surveillance. Across a number of infectious diseases there is now a concerted effort towards integrated sero-surveillance, where samples from the same cross-sectional survey are tested for antibody responses to multiple pathogens simultaneously, using a multiplexed bead-based immunoassay platform[31,32]. Given the enormous potential of integrated surveillance afforded by multiplexing, it is likely that an increasing amount of data of the format analysed here will become available in the future. This will increase the need for appropriate analytic methods. The approaches applied here could be adopted for other diseases, to help generate understanding of low-transmission scenarios, determine how such data can complement existing programme evaluation methods and guide future study design in the post-MDA phase.

## Methods

**Data**. We used the data on antibody responses and TF collected from trachoma-endemic populations of Fiji (two different ethnic populations), Kiribati, Nepal, Solomon Islands, The Gambia, and the United Republic of Tanzania (Rombo district). A map which highlights the countries from which data are available is presented in the Supplementary Information (Supplementary Figure 2). Full details of the field and laboratory methodologies, relevant ethical and regulatory board approval are provided elsewhere[14,16,33–35]. A summary of the demography of each population is provided in Supplementary Table 1, and age-specific prevalence curves for TF and sero-positivity to $Ct$ antigens are presented in Fig. 1. Data on antibody response against Pgp3 and CT694 were available for Nepal and Rombo, whilst serological data from all other sites were limited to Pgp3. Thresholds to determine sero-positivity were defined for each dataset separately. Data generated using multiplex bead array (Nepal and Rombo), were provided as median fluorescence intensity (MFI) with background subtracted out (MFI-BG), with threshold cut-offs determined using a receiver operating characteristic (ROC) curve, based on previously-assayed dried blood spots in children in a trachoma-endemic district of Tanzania (Kongwa district)[7]. Thresholds for the data collected as optical density (OD) values were defined as the mean plus three SDs of the OD in sero-negative controls[36].

**Ethics**. As a secondary analysis of pre-existing data sets, this study was considered by the Research Ethics Review Committee of the World Health Organization to be exempt from full review (0003001).

**Modelling and model fitting**. To investigate age and exposure-dependent changes in anti-Pgp3 and anti-CT694 antibody responses, data were analysed using two distinct statistical models: (i) *sero-catalytic models* to describe how the proportion of sero-positive individuals changes with age, dependent on historical exposure to $Ct$[8,9]; and (ii) *antibody acquisition models* to describe how geometric mean antibody levels change with age and exposure[17]. Age-dependent variation in sero-prevalence or antibody levels can be used to infer historical patterns of trachoma transmission. For both sero-catalytic and antibody acquisition models, three distinct transmission scenarios were considered: scenario 1 assumed a constant rate of transmission; scenario 2 assumed a sharp drop in transmission $t_c$ years ago; and scenario 3 assumed a linear decline in transmission[8,17]. Assumptions and parameters estimated from the data for each scenario are presented in Table 1. Full mathematical descriptions of the models are provided in the Supplementary Methods.

A key challenge when modelling serological data is parameter identifiability, where the data may not provide sufficient statistical power to allow precise parameter estimation. An important example occurs when estimating SCRs ($\lambda$) and SRRs ($\rho$). If the data are limited, it is not always possible to distinguish between the case where antibodies are acquired rapidly and then decay rapidly, and the case where antibodies are acquired slowly and decay slowly[37]. However, the cohorts in this study were considered of sufficient size to allow estimation of sero-catalytic model parameters[37]. We tested variants of each model while fixing the SRR with the estimate from Nepal. Antibodies to Pgp3 and CT694 are also generated during urogenital $Ct$ infection, therefore, we accounted for a second potential source of antigen exposure following sexual debut at age 18 years[28]. We fitted up to 10 different sero-catalytic models to each dataset for which the data on the whole population were available, and six different models where information on individuals <16 years of age were available. For both the sero-catalytic and antibody acquisition models, inference was implemented at population level, and we therefore consider sero-conversion, reversion and antibody acquisition and decay at population level.

Parameter estimation was performed using Markov Chain Monte Carlo, and chain convergence was assessed by ensuring a Gelman-Rubin statistic <1.1[38] and effective sample size >350. Model comparison statistics were calculated using the DIC[39]. Although historical exposure to trachoma can determine age-stratified sero-prevalence curves, serological data from a single cross-sectional study are often insufficient to statistically distinguish between competing explanations for changes in transmission intensity. If several scenarios provided comparable descriptions of the data (as judged by DIC), we selected the scenario that best fitted with prior epidemiological knowledge of that setting, e.g. the occurrence of an intervention program, or the transmission of urogenital *Chlamydia*. Additional details on the modelling methodology, fitting and convergence diagnostics are provided in the Supplementary Methods. All calculations and analysis were performed using R version 3.3.1[40].

We compared results from the sero-prevalence models across all sites. However, due to the use of different antibody assays in different sites, results from antibody acquisition models could not be compared between sites. To understand the relationship between different measures of transmission intensity (SCR and TF prevalence) we fitted a linear model to the relationship between $\lambda_T$ (for Pgp3) and TF prevalence for each study site. We fitted $\lambda_T$ as a function of TF prevalence, and predicted the relationship between TF prevalence and $\lambda_T$.

**Code availability**. All code is available on GitHub: https://github.com/Pinzo1/Serology_code-.

## Data availability

All data analysed in this study are previously published. The authors declare that all other data supporting the findings of this study are available within https://github.com/Pinzo1/Serology_code-, or are available from the authors upon request.

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

## Acknowledgements

The authors alone are responsible for the views expressed in this article and they do not necessarily represent the views, decisions or policies of the institutions with which they are affiliated. Funding for collection of samples from Nepal and Rombo was provided by the U.S. Agency for International Development under an Interagency Agreement with CDC. A.P. gratefully acknowledges funding of the NTD Modelling Consortium by the Bill and Melinda Gates Foundation in partnership with the Task Force for Global Health. The views, opinions, assumptions or any other information set out in this article should not be attributed to Bill & Melinda Gates Foundation and The Task Force for Global Health or any person connected with them. We thank Chrissy Roberts, Robert Butcher and Michael Marks of London School of Hygiene & Tropical Medicine for original data collection and helpful discussions. The authors are especially grateful to the study participants and field teams.

## Author contributions

A.P. and M.T.W.: conceived and designed the study, performed the analysis, interpreted the results and wrote the first draft of the manuscript. A.W.S. and D.L.M.: interpreted the results and wrote the first draft of the manuscript. R.L.B., R.B., A.C., D.D., B.G., S.G., K.R.J., R.P.K., M.K., P.M., C.M., D.C.W.M., S.M., A.M., F.S., O.S., R.Taoaba, R.Tekeraoi: conceived and designed the epidemiological studies that generated the data analysed in the study. All authors read and approved the final manuscript.

## Additional information

**Competing interests:** The authors declare no competing interests.

