## [Peer Review File · Nature Communications]

Reviewers' Comments:

Reviewer #1:

Remarks to the Author:

In this manuscript, Pinsent and colleagues use mathematical models to analyze serologic data from trachoma surveys in multiple countries to understand transmission patterns. The results suggest that serologic data may be useful for trachoma surveillance.

I have only a few comments.

1) It is encouraging to see that antibody prevalence may have a role as an objective indicator of trachoma transmission. In terms of documenting that antibody prevalence is less than a designated threshold, why didn't the authors mention LQAS sampling as the basis of the survey methodology? As with the LF transmission assessment survey, could antibody prevalence be assessed in a more narrowly defined age group?

2) If serologic assays were to be used programmatically, is it realistic to think that lab based serologic assays could be used for this purpose?

3) I'm trying to understand why a short-lived bacterial infection is associated with such a long antibody half-life. Is there any reason to think that this represents tissue persistence?

Editorial Comments:

1) The abstract seems a bit disjointed to me.

2) Line 52 - "probelm" should be "problem"

3) Strictly speaking, antibody assays conducted using a single dilution of serum should be referred to as a "level" rather than as a titer.

Reviewer #2:

Remarks to the Author:

This is an article that tries to do two things. 1st, it's analyzing a lot of serology data for Chlamydia trachomatis, which causes trachoma. 2nd, it's making an argument that this is worth doing because of limitations and non-linearities from other sorts of data. Overall, the analysis is solid. It uses methods that are well-accepted in malaria and some other fields, which face many of the same problems in measuring transmission. In malaria, these are now well-accepted analytical methods with known limitations, and this is becoming pretty standard stuff. I think the paper would be a solid contribution to the literature in more or less its present form, and it would represent an important contribution to the trachoma. The methods and diagnostics are statistically sound and well-explained in the supplement. I don't have too many critical things to say about substance.

My major recommendation is primarily cosmetic -- the way the concepts and results are presented.

The first sentence of the abstract -- putting the "non-linear" bit first -- is bad salesmanship. The problem here is to measure transmission of a pathogen -- and that should be front and center. The non-linearities are an obvious problem, but the serology solves the problem of measuring

transmission. The non-linearities are secondary (and crucially important). I would open the abstract with something close to the line on page 3, line 55. "Robust surveillance methods are needed for trachoma control" ... but "existing methods have limitations."

At the end of the last paragraph on page 3, I would say directly that there is a need for methods that work at very low transmission, and that serology is one option.

Beginning of next paragraph, say "for trachoma and numerous other infectious diseases." not "can quantify" but "have been used to measure"

The background should do a better job of putting this analysis into context. In particular, it would be worth pulling the discussion about previous analysis from page 8 line 216 up into the introduction to put your study into context).

The first paragraph of the discussion should do a better job of emphasizing what you found, not what you did. Our analysis shows that....

I re-emphasize that these are mostly cosmetic changes to the text in the background and discussion that I believe would strengthen the paper. The analysis is very well explained and solid.

Reviewer #3:

Remarks to the Author:

This is an important and well-written paper about the usefulness of serological surveys in trachoma to understand exposure history, changes in transmission, and its ability to inform current thresholds for elimination of trachoma as a public health problem.

The modelling is appropriate and well described. The results are well discussed and the limitations of the study are acknowledged and well explained. Suggestions for further work are given.

I have annotated the paper with my comments and attach the annotated MS to this review.

The following is a summary of my comments:

- 1) MDA is a generic name that should be spelled out and also the drugs used in MDA should be named; I assume that azithromycin will be the main one.
- 2) Some abbreviations without explanations are given in the Abstract.
- 3) In the Abstract it should be highlighted that the TF threshold is for trachoma elimination as a public health problem.
- 4) Since the Methods are at the end, some brief explanations should be given in the Results to avoid the reader having to refer constantly to the Methods section.
- 5) Perhaps Table 1 should be expanded to describe the datasets a bit better and to give some background to the epidemiology and history of trachoma interventions in the settings investigated.
- 6) I also wonder if a map could be added to show the geographical spread of the datasets.
- 7) Re-state as appropriate that elimination refers to elimination as a public health problem.

Reviewer #4:

Remarks to the Author:

The manuscript entitled "The utility of serology for elimination surveillance: an example from

trachoma" embodies an interesting trachoma modelling work and a potentially useful tool for verification of trachoma elimination from the trachoma endemic areas in the world. I believe it makes an important contribution to the trachoma elimination modelling, in particular, and to the NTD modelling enterprise, in general. Having said that, however, I find very difficult to make a recommendation for its publication at Nature Communications without major revisions of the manuscript. The main reason for my decision is based on my evaluation of the manuscript as I describe in what follows.

First, although I like the title of the manuscript, the authors' motivation behind the project lacks in enthusiasm in the introductory section. (No compelling case has been made: for example, if the trachoma serology modelling is not done and this "proposed" tool is not developed, how bad would it be for the ongoing trachoma elimination programmes around the world?)

I did not find the abstract providing me enough information for reading the manuscript any further, i.e., beyond the end of the abstract. Why? Although the very first sentence in Abstract is clumsily written (would it not have been sufficient to say that "The non-linear relationship between infection and disease prevalence is challenging for surveillance."?), the first 2 sentences tell readers as to why the authors did what they did (mentioned in the following 3rd sentence) in this project. Overall, I did not find the abstract capturing the essence of the work carried out and the results presented in the manuscript.

Now, let us take a closer look at the first few paragraphs of the background section. The first two paragraphs seemingly sound very customary for a standard trachoma modelling paper. (By a standard trachoma modelling paper I mean that the results and conclusion derived/discussed in it are relevant for trachoma.) Personally I find that the sentence at line 65 really starts what [I think, the authors would have ideally started with, in line with the title of the manuscript] for readers to get engaged in reading the manuscript as a modelling work, which, again as per my opinion, could initiate and/or invigorate a fresh thinking about elimination modelling beyond trachoma.

Second, very poor presentation of the results. Most of the figure captions poorly describe what potential readers (certainly for myself) would have taken away by reading the [properly described] captions and having a look at the plots in the figures. The situation becomes worse when a figure is comprised of a number of panels. Take Figure 1 as an example. Where are grey polygons in the panels? I see grey lines. What are grey triangles? Why do I not see any plot legends? In addition to similar examples (throughout the relevant figures - the figures with several panels - in the supplementary material), why did the authors not bother describing the panel (b) in the caption of Figure 3. These (and many others, which I am not going to present here individually, throughout the text and the supplementary material) show how badly the authors have done the proofreading of the final draft.

Here are a few examples of bad proofreading.

Take Figure S1. I am not able to see what the numbers on x- and y-axes are. Neither can I read the y-labels. One thing I would like point out here. If all panel plots in a row of a multi-panel figure have the same y-labels, why not give the y-label only on the left-most panel plot in that row. This will improve the readability.

The name Gambia is preceded with "the." There are several instances in the text and in the supplementary material, where Gambia is used without the.

The authors chose past sentence to describe the methods. But on p 4 in Model fitting and comparison

of the supplementary material, the authors liberally use both the past and present sentences. Please stick with one.

Third, the mathematical description of the models is not clear.

The following I copied and pasted from p 1 of the supplementary material.

"For each model, we denote t to be the time before the transmission period; this differed for each data set and was calculated as the date at which samples were collected minus the age of the oldest individual in the dataset. t_{end} was the time at the end of the study period and hence the time at which the most recent cross-section was collected."

It is not clear what the transmission period the authors mean to denote? Also, does t take negative values? t_{end} is the same across all sites. Does that mean the simulation time period varied for the models across the sites? It is so confusing.

Why did the authors not provide the analytic expressions of $P(a)$ for Scenarios 1 and 2 for the models for seroconversion with urogenital exposure in the supplementary material?

Ditto for all 3 scenarios for the models for antibody acquisition with urogenital exposure?

I encourage the authors to do so. In doing so, it would be much help if some non-trivial steps, where applicable, are also provided.

How an effective size of >350 was arrived at? Any justification for using less or more than this cutoff value?

Why the posterior distributions of the parameters of the best selected model for a study site is not shown in the supplementary material?

Lastly, I do not see the final take away message, clearly spelled out in the discussion section.

In addition, the authors say "modelling suggested transmission of trachoma had declined over time" (at line 183 in the text). How one could verify this statement in the field? And which sites (with which intervention characteristics, such as, high MDA coverage, etc.) indicated better decline than the others?

The last paragraph (lines 269 - 276) of the discussion section is very general. In the sentence "... here we have demonstrated the potential utility of antibody based surveillance to help monitor infection recrudescence after validation.", what does validation stand for? It is neither clear how one should design antibody data collection in the post-MDA phase of elimination programmes.

The authors say that at line 341 "All code is available to editors and referees upon request."

My comment is that the computer codes used, along with any input files, should be made available to editors and reviewers without their request in order for facilitating a rigorous review process.

Response to reviewer comments for the manuscript entitled: The utility of serology for elimination surveillance: an example from trachoma - NCOMMS-18-15959

We thank all 4 reviewers for the helpful and insightful comments they have provided on the manuscript. We have reproduced all reviewer comments in **bold** and our point by point response to each comment is provided in *italics*.

Reviewer #1

General comments

In this manuscript, Pinsent and colleagues use mathematical models to analyze serologic data from trachoma surveys in multiple countries to understand transmission patterns. The results suggest that serologic data may be useful for trachoma surveillance.

Specific comments

I have only a few comments.

[1.1] It is encouraging to see that antibody prevalence may have a role as an objective indicator of trachoma transmission. In terms of documenting that antibody prevalence is less than a designated threshold, why didn't the authors mention LQAS sampling as the basis of the survey methodology? As with the LF transmission assessment survey, could antibody prevalence be assessed in a more narrowly defined age group?

[Response 1.1] We thank the reviewer for raising this very interesting point. Currently sero-surveillance for trachoma is in its infancy. Within our study we have tried to be as epidemiologically robust as possible and include maximally wide age ranges because we firstly need to develop our understanding of the science relating to the antibody dynamics as a result of Ct exposure. Areas requiring greater exploration include: the duration of sero-positivity after seroconversion, and how findings from serological data relate to those from more established programme monitoring tools. After this has been done it may be possible to develop a framework for trachoma which has many conceptual similarities to LQAS sampling used for lymphatic filariasis. However, we believe that consideration of how serosurveillance could be operationalised is a step for future studies once a robust understanding of the science has been generated.

[1.2] If serologic assays were to be used programmatically, is it realistic to think that lab based serologic assays could be used for this purpose?

[Response 1.2] We thank the reviewer for raising this important point. Across a number of infectious diseases there is now a concerted effort towards integrated sero-surveillance, where samples from the same cross-sectional survey are tested for antibody responses to multiple pathogens simultaneously, using a multiplex bead-based immunoassay platform such as Luminex. Therefore, given the enormous potential of integrated surveillance afforded by multiplexing, it is likely that an increasing amount of data of the format analysed here will become available, along with a concomitant need for appropriate analytic methods. Thus we feel that serological assays will be and could be used for this purpose. On pages 10-11, lines 298-310 of the discussion we now say the following:

“However, here we have demonstrated the potential utility of antibody-based surveillance to help monitor low-levels of ocular Ct infection transmission. We have initiated an evidence

base for the use of sero-surveillance by programmes in low-transmission and post-elimination settings (where TF <5%), and have provided an operational threshold for sero-surveillance. Across a number of infectious diseases there is now a concerted effort towards integrated sero-surveillance, where samples from the same cross-sectional survey are tested for antibody responses to multiple pathogens simultaneously, using a multiplexed bead-based immunoassay platform^{34,35}. Given the enormous potential of integrated surveillance afforded by multiplexing, it is likely that an increasing amount of data of the format analysed here will become available in the future. This will increase the need for appropriate analytic methods. The approaches applied here could be adopted for other diseases, to help generate understanding of low-transmission scenarios, determine how such data can complement existing programme evaluation methods and guide future study design in the post-MDA phase.”

[1.3] I’m trying to understand why a short-lived bacterial infection is associated with such a long antibody half-life. Is there any reason to think that this represents tissue persistence?

[Response 1.3] We have clarified this in the discussion of the manuscript. While it is impossible to definitely rule out bacterial persistence in tissue even after treatment, the most plausible explanation is that even short-lived infections can induce populations of long-lived plasma cells which secrete antibodies for many years after the initial infection. We highlight this in the manuscript on page 8, lines 212-219 where we now say the following:

“In most settings, the sero-prevalence of anti-Ct antibodies increased with age. A typical ocular Ct infection commonly lasts only a few weeks or months, which may not be sufficient to develop long-lived plasma cells that secrete antibodies for many years, even in the absence of subsequent infection that would provide “boosting” of the response. Though it is impossible to rule out bacterial persistence in tissues, it is more likely that the increase in sero-prevalence with increasing age is due to repeated infection with ocular Ct leading to the induction of plasma cells²¹⁻²³. Therefore, we believe that the measured response reflects cumulative exposure to the pathogen and immunological memory.”

[1.4] The abstract seems a bit disjointed to me.

[Response 1.4] We have taken this comment on board and have amended the introduction to the abstract, as this was also a comment pointed out by reviewer 2. We have now modified the motivation for our study and re-focused the start of the abstract, on page 3, lines 37-40 we now say the following:

“Robust surveillance methods are needed for trachoma control and recrudescence monitoring, but existing methods have limitations. Sero-surveillance is being considered as an additional tool to address issues associated with current approaches. We analysed data from nine settings, assessing the potential programmatic contribution of serological data.”

[1.6] Line 52 - “probelm” should be “problem”

[Response 1.6] We thank the reviewer for highlighting this typo, we have now corrected it. This can be found on page 4, line 100.

[1.7] Strictly speaking, antibody assays conducted using a single dilution of serum should be referred to as a “level” rather than as a titer.

[Response 1.7] *We appreciate this comment from the reviewer and thank them for highlighting this very valid point. We have taken this comment on board and have now changed any reference of 'antibody titre' to 'antibody level' in the manuscript and all figure legends where applicable.*

Reviewer #2

General comments

[2.1] **This is an article that tries to do two things. 1st, it's analyzing a lot of serology data for *Chlamydia trachomatis*, which causes trachoma. 2nd, it's making an argument that this is worth doing because of limitations and non-linearities from other sorts of data. Overall, the analysis is solid. It uses methods that are well-accepted in malaria and some other fields, which face many of the same problems in measuring transmission. In malaria, these are now well-accepted analytical methods with known limitations, and this is becoming pretty standard stuff. I think the paper would be a solid contribution to the literature in more or less its present form, and it would represent an important contribution to the trachoma. The methods and diagnostics are statistically sound and well-explained in the supplement. I don't have too many critical things to say about substance.**

[Response 2.1] *We greatly appreciate the general comments provided on the manuscript by the reviewer, and we are delighted that they believe our article will make a strong contribution to the research field.*

Specific comments

My major recommendation is primarily cosmetic -- the way the concepts and results are presented.

[2.2] **The first sentence of the abstract -- putting the "non-linear" bit first -- is bad salesmanship. The problem here is to measure transmission of a pathogen -- and that should be front and center. The non-linearities are an obvious problem, but the serology solves the problem of measuring transmission. The non-linearities are secondary (and crucially important). I would open the abstract with something close to the line on page 3, line 55. "Robust surveillance methods are needed for trachoma control" ... but "existing methods have limitations."**

[Response 2.2] *We thank the reviewer for this very helpful suggestion. We have amended the abstract in accordance with the recommendations provided, and feel this has improved the flow of the abstract. On page 3, lines 37-40 we now say the following:*

"Robust surveillance methods are needed for trachoma control and recrudescence monitoring, but existing methods have limitations. Sero-surveillance is being considered as an additional tool to address issues associated with current approaches. We analysed data from nine settings, assessing the potential programmatic contribution of serological data."

[2.3] **At the end of the last paragraph on page 3, I would say directly that there is a need for methods that work at very low transmission, and that serology is one option.**

[Response 2.3] *We thank the reviewer for this helpful suggestion. We have added in the additional suggested sentence and on page 3, lines 67-68 we now say the following:*

“There is therefore a clear need for surveillance methods that accurately monitor low levels of transmission, and serology is potentially one such method.”

[2.4] Beginning of next paragraph, say "for trachoma and numerous other infectious diseases." not "can quantify" but "have been used to measure"

[Response 2.4] *We thank the reviewer for this helpful comment, we have now amended the sentence as per the reviewers recommendation, on page 4, lines 71-72 we now say the following:*

“antibody responses resulting from a single or cumulative exposure to a pathogen have been used to measure and assess changes in transmission intensity”

[2.5] The background should do a better job of putting this analysis into context. In particular, it would be worth pulling the discussion about previous analysis from page 8 line 216 up into the introduction to put your study into context).

[Response 2.5] *We thank the reviewer for this recommendation. We have now provided more context and discussion on previous analysis and what future work needs to be done in the introduction of our manuscript. On page 4, lines 90-100 we now say the following:*

“The collection and analysis of Ct serology data is an ongoing and active area of trachoma research. A previous modelling analysis of serological data from Rombo, Tanzania suggested that a step-wise drop in transmission occurred ~15 years prior to the survey date¹⁶. Equally, a study analysing data from The Gambia also suggested that a step-wise drop in transmission occurred 19-23 years prior to sampling¹⁷. However, in analyses to date, no consideration has been given to age-dependent exposure to urogenital Ct when estimating the sero-conversion rate (SCR). Furthermore, previous studies have either assumed that sero-reversion following conversion does not occur at all¹⁶ or that it takes on average at least 65 years¹⁷ and estimates of this parameter have been limited to data from one cross-sectional survey. Therefore more research is required in order to estimate the sero-reversion rate from multiple cross-sectional surveys, and to estimate the SCR whilst accounting for the potential exposure to urogenital Ct in settings where this may be a problem.”

[2.6] The first paragraph of the discussion should do a better job of emphasizing what you found, not what you did. Our analysis shows that....

[Response 2.6] *We thank the reviewer for this constructive comment. We have re-written the first paragraph of the discussion to highlight clearly the key findings from our analysis. On page 8, lines 200-211 we now say the following:*

“Serological surveillance for trachoma is being considered to help programmes undertake post-validation surveillance. Prior analyses of the potential utility of serology have used only single-study-site data, with few studies taking a quantitative modelling approach^{14,19,20}. In all settings evaluated, our modelling suggested transmission of trachoma had declined over time, either through a step-change in transmission, or more linearly. We estimated SRR half-life to be 26 (95% CrI: 21-34) years and 40 (95% CrI: 33-53) years, with an estimated half-life of the antibody response to be 7.3 (95% CrI: 6.5-8.2) years and 5.5 (95% CrI: 4.7-6.3) years, for Pgp3 and CT694 respectively: the first estimates published for these parameters. Our results suggested that SCRs below 0.015 year⁻¹ correspond to TF <5%, and that the mean sero-prevalence for 1–9-year-olds when TF <5% was <7%. Whilst more data are needed to reduce the uncertainty in the relationship between sero-prevalence and TF prevalence, we

present an initial operational threshold for interpreting serological data in trachoma programmes.”

[2.7] I re-emphasize that these are mostly cosmetic changes to the text in the background and discussion that I believe would strengthen the paper. The analysis is very well explained and solid.

[Response 2.7] We thank the reviewer for all of the helpful comments provided and their supportive review of the manuscript.

Reviewer #3

General comments

[3.1] This is an important and well-written paper about the usefulness of serological surveys in trachoma to understand exposure history, changes in transmission, and its ability to inform current thresholds for elimination of trachoma as a public health problem. The modelling is appropriate and well described. The results are well discussed and the limitations of the study are acknowledged and well explained. Suggestions for further work are given. I have annotated the paper with my comments and attach the annotated MS to this review.

[Response 3.1] We thank the reviewer for their insightful comments and supportive review of our manuscript. We feel following their recommendations has helped to improve the quality and clarity of the manuscript.

Specific comments

The following is a summary of my comments:

[3.2] MDA is a generic name that should be spelled out and also the drugs used in MDA should be named; I assume that azithromycin will be the main one.

[Response 3.2] We thank the reviewer for raising this to our attention. We have now defined MDA before its first use in the text and stated that azithromycin is the drug that is distributed. This clarification can be found on page 3, lines 57-59, we now say the following:

“specifically, for the number of rounds of azithromycin mass drug administration (MDA) rounds required before re-estimation of prevalence”

[3.3] Some abbreviations without explanations are given in the Abstract.

[Response 3.3] We thank the reviewer for highlighting this. We have now defined both SCR and TF in the abstract, this can be found on page 3, lines 44-46, we now say the following:

“Estimates of the sero-conversion rate (SCR) below 0.015 (95% Confidence Interval (CI): 0.0-0.049) year⁻¹ corresponded to a prevalence of trachomatous inflammation-follicular (TF) <5%.”

[3.4] In the Abstract it should be highlighted that the TF threshold is for trachoma elimination as a public health problem.

[Response 3.4] We certainly agree with the reviewer and thank them for raising this to our attention, we have now clarified this on page 3, lines 46-47. We now say the following:

“corresponded to a prevalence of trachomatous inflammation-follicular (TF) <5% (the current threshold for the elimination of trachoma as a public health problem)”

[3.5] Since the Methods are at the end, some brief explanations should be given in the Results to avoid the reader having to refer constantly to the Methods section.

[Response 3.5] *We thank the reviewer for this helpful suggestion. We have now added in an additional paragraph at the start of the results section which provides a brief overview on our methods to aide readability of the manuscript. On page 5, lines 106-116 we now say the following:*

“We performed analysis on 9 datasets from 6 different geographic regions and analysed the data using two different statistical models: sero-catalytic models and antibody acquisition models. We evaluated age-dependent changes in anti-Pgp3 and anti-CT694 antibody prevalence to infer historical patterns of transmission within each setting. For each of the different model types (sero-prevalence and antibody acquisition) three distinct transmission scenarios were considered: scenario 1 assumed a constant rate of transmission; scenario 2 assumed a sharp drop in transmission t_c years ago; and scenario 3 assumed a linear decline in transmission^{8,18}. We fitted up to 10 different transmission scenarios to each dataset. To understand the relationship between different measures of transmission intensity (SCR and TF prevalence) we fitted a linear model to the relationship between the SCR for trachoma (λ_T) (for Pgp3) and TF prevalence for each study site. Full details on the methodology are provided in the Methods section and Supplementary information.”

[3.6] Perhaps Table 1 should be expanded to describe the datasets a bit better and to give some background to the epidemiology and history if trachoma interventions in the settings investigated.

[Response 3.6] *We thank the reviewer for this suggestion. We have taken this on board and provided a number of additional columns to Table 2 to provide more information on the demography of the different study populations. We have now added a column for age range, total sero-prevalence and the total number of samples analysed. These changes can now be seen in Table 2. We note that age-group specific demography data and TF prevalence is provided in Table S2 of the Supplementary information.*

[3.7] I also wonder if a map could be added to show the geographical spread of the datasets.

[Response 3.7] *We have now presented a map in the supplementary information which highlights in red the regions of the world from which data were analysed. This figure can be found on page XX of the supplementary information and is referenced to in the text on page 11, lines 315-316, we now say:*

“A map which highlights the countries from which data were available is presented in the Supplementary information (Figure S2).”

[3.8] Re-state as appropriate that elimination refers to elimination as a public health problem.

[Response 3.8] *We thank the reviewer for making this valid point, we have propagated this edit throughout the manuscript to ensure clarity for the readership.*

Reviewer #4

General comments

The manuscript entitled "The utility of serology for elimination surveillance: an example from trachoma" embodies an interesting trachoma modelling work and a potentially useful tool for verification of trachoma elimination from the trachoma endemic areas in the world. I believe it makes an important contribution to the trachoma elimination modelling, in particular, and to the NTD modelling enterprise, in general. Having said that, however, I find very difficult to make a recommendation for its publication at Nature Communications without major revisions of the manuscript. The main reason for my decision is based on my evaluation of the manuscript as I describe in what follows.

Specific comments

[4.1] First, although I like the title of the manuscript, the authors' motivation behind the project lacks in enthusiasm in the introductory section. (No compelling case has been made: for example, if the trachoma serology modelling is not done and this "proposed" tool is not developed, how bad would it be for the ongoing trachoma elimination programmes around the world?)

[Response 4.1] *The aim of this project i.e modelling serological data is motivated by a number of factors. Firstly, studies are typically reporting sero-surveillance from one site and often only 1 cross-section, this means that no collective analysis of multiple serological datasets has been performed previously, this is what we present in the current manuscript. Before any form of policy recommendations can be provided on the use of serology for trachoma or any other disease it needs to be understood how findings from serological data may relate to findings about transmission obtained from current standard tools, we present a framework for how this may be done in this article. Furthermore, existing modelling work of serological data has been limited to individual study sites and has not accounted for important factors such as the possibility of urogenital chlamydia transmission within a region or allowed for sero-reversion or been able to estimate this important parameter from multiple epidemiological cross-sections collected from the same study region. Sero-surveillance is being evaluated by WHO as a potential post-validation monitoring tool, it is therefore important to understand what the collection of this serological data can tell us about trachoma transmission. In the introduction we have provided some additional background information on previous serological work for trachoma to motivate why our study is important and has been conducted. On page 4, lines 90-100 we now say the following:*

"The collection and analysis of Ct serology data is an ongoing and active area of trachoma research. A previous modelling analysis of serological data from Rombo, Tanzania suggested that a step-wise drop in transmission occurred ~15 years prior to the survey date¹⁶. Equally, a study analysing data from The Gambia also suggested that a step-wise drop in transmission occurred 19-23 years prior to sampling¹⁷. However, in analyses to date, no consideration has been given to age-dependent exposure to urogenital Ct when estimating the sero-conversion rate (SCR). Furthermore, previous studies have either assumed that sero-reversion following conversion does not occur at all¹⁶ or that it takes on average at least 65 years¹⁷ and estimates of this parameter have been limited to data from one cross-sectional survey. Therefore more research is required in order to estimate the sero-reversion rate from multiple cross-sectional surveys, and to estimate the SCR whilst accounting for the potential exposure to urogenital Ct in settings where this may be a problem."

[4.2] I did not find the abstract providing me enough information for reading the manuscript any further, i.e., beyond the end of the abstract. Why? Although the very first sentence in Abstract is clumsily written (would it not have been sufficient to say that "The non-linear relationship between infection and disease prevalence is challenging for surveillance."?), the first 2 sentences tell readers as to why the authors did what they did (mentioned in the following 3rd sentence) in

this project. Overall, I did not find the abstract capturing the essence of the work carried out and the results presented in the manuscript.

[Response 4.2] *We have now modified the abstract, and with the amended changes suggested and provided by reviewer 2 we now feel that the abstract is more captivating. On page 3, lines 37-48 we now say the following:*

“Robust surveillance methods are needed for trachoma control and recrudescence monitoring, but existing methods have limitations. Sero-surveillance is being considered as an additional tool to address issues associated with current approaches. We analysed data from nine settings, assessing the potential programmatic contribution of serological data. Serology provided insight into the transmission history within each population. For study sites with high sero-prevalence in adults, it was necessary to account for secondary exposure to Ct antigens due to urogenital infection. We estimated the population half-life of sero-reversion to be 26 (95% Credible Interval (CrI): 21-34) years. Estimates of the sero-conversion rate (SCR) below 0.015 (95% Confidence Interval (CI): 0.0-0.049) year⁻¹ corresponded to a prevalence of trachomatous inflammation—follicular (TF) <5% (the current threshold for the elimination of active trachoma as a public health problem). As global trachoma prevalence declines, we may need cross-sectional serological survey data to inform programmatic decisions.”

[4.3] Now, let us take a closer look at the first few paragraphs of the background section. The first two paragraphs seemingly sound very customary for a standard trachoma modelling paper. (By a standard trachoma modelling paper I mean that the results and conclusion derived/discussed in it are relevant for trachoma.) Personally I find that the sentence at line 65 really starts what [I think, the authors would have ideally started with, in line with the title of the manuscript] for readers to get engaged in reading the manuscript as a modelling work, which, again as per my opinion, could initiate and/or invigorate a fresh thinking about elimination modelling beyond trachoma.

[Response 4.3] *We appreciate this comment made by the reviewer however, we feel that it is necessary and important to provide a general introduction to trachoma as we're targeting the broader NTD community, who may not necessarily be familiar with the basics of trachoma epidemiology and elimination targets. We have therefore chosen to keep this background information in.*

[4.4] Second, very poor presentation of the results. Most of the figure captions poorly describe what potential readers (certainly for myself) would have taken away by reading the [properly described] captions and having a look at the plots in the figures. The situation becomes worse when a figure is comprised of a number of panels. Take Figure 1 as an example. Where are grey polygons in the panels? I see grey lines. What are grey triangles? Why do I not see any plot legends? In addition to similar examples (throughout the relevant figures - the figures with several panels - in the supplementary material), why did the authors not bother describing the panel (b) in the caption of Figure 3. These (and many others, which I am not going to present here individually, throughout the text and the supplementary material) show how badly the authors have done the proofreading of the final draft.

[Response 4.4] *We have now amended the legends for clarity. Grey polygons refer to the shaded area about the median prediction, we have now simply referred to this as the grey shaded area. For the legend for Figure 1 on page 17, lines 517-526 we now say the following:*

“Figure 1. Fits of the best performing sero-catalytic models to age-specific sero-prevalence data for each of the 9 study sites for both antigens. The titles within each panel indicate the study site, antigen-specific antibody responses measured and the best fitting transmission scenario for that dataset. Black squares indicate the proportion sero-positive in each age-group and green triangles indicate the age-group specific TF prevalence. Black and green data points on the Nepal plots indicate pre and post-MDA respectively. Error bars on the squares and triangles indicate binomial confidence intervals. Solid black lines running through the sero-prevalence data were generated with the median parameter estimates from each model fit. The shaded grey region represents the 95% credible intervals of the model predictions. Uncertainty was generated by drawing 500 independent samples from the posterior distribution.”

We have also added additional detail to the legend of Figure 3, on page 17, lines 534-543 we now say the following:

“Figure 3. The estimated relationship between the sero-conversion rate (SCR) and TF prevalence and the predicted proportion of people sero-positive for a given level of TF prevalence determined using the relationship estimated from the linear model. (a) Black dots indicate the median estimated SCR for each dataset and the TF prevalence from each of the 9 study sites. The solid black line is the mean predicted relationship between the SCR and TF prevalence, obtained by fitting a linear model to the data. The 95% confidence intervals about the mean relationship are indicated as grey dashed lines. (b) the predicted mean proportion of people sero-positive for a given level of TF prevalence is shown with a solid black line, the 95% confidence intervals about this mean are indicated with dashed grey lines. For the elimination as a public health problem threshold of TF <5%, we would expect 6.2% (95% CI: 0.0%-19.9%) to test sero-positive.”

[4.5] Here are a few examples of bad proofreading. Take Figure S1. I am not able to see what the numbers on x- and y-axes are. Neither can I read the y-labels. One thing I would like point out here. If all panel plots in a row of a multi-panel figure have the same y-labels, why not give the y-label only on the left-most panel plot in that row. This will improve the readability.

[Response 4.5] We have now increased the font size on all figures in the supplementary information. In addition we have also included more detail in the legends of each of the supplementary figures to ensure that presentation of the results is clear. We have kept the axis labels of each figure as they are and just increased the size of the labels to aid readability, we feel that changing from 1 to multiple axis labels is a comment pertaining primarily to style and taste, and was not noted by any other reviewers as a necessary specific change, we have therefore left them in place.

[4.6] The name Gambia is preceded with "the." There are several instances in the text and in the supplementary material, where Gambia is used without the.

[Response 4.6] We thank the reviewer for bringing this to our attention. We have been through the manuscript and Supplementary information to correct all instances of this.

[4.7] The authors chose past sentence to describe the methods. But on p 4 in Model fitting and comparison of the supplementary material, the authors liberally use both the past and present sentences. Please stick with one.

[Response 4.7] We thank the reviewer for raising this. We have thoroughly proof-read this section and ensured that past tense is now consistently used throughout this section. These changes can now

be found on page 4 of the supplementary information.

[4.8] **Third, the mathematical description of the models is not clear. The following I copied and pasted from p 1 of the supplementary material. "For each model, we denote t to be the time before the transmission period; this differed for each data set and was calculated as the date at which samples were collected minus the age of the oldest individual in the dataset. t_{end} was the time at the end of the study period and hence the time at which the most recent cross-section was collected." It is not clear what the transmission period the authors mean to denote? Also, does t take negative values? t_{end} is the same across all sites. Does that mean the simulation time period varied for the models across the sites? It is so confusing.**

[Response 4.8] *We thank the reviewer for this helpful comment. We have now rephrased these sentences in the supplementary information for clarity, and on page 1, we now say the following instead:*

"For each model, we used t to denote time. t_{end} was the time that the most recent cross-sectional survey occurred (i.e. the last date of fieldwork). t_{start} was the year in which the oldest person at the first cross-sectional survey was born. The maximum transmission period evaluated was thus the difference in years between t_{end} and t_{start} ."

[4.9] **Why did the authors not provide the analytic expressions of $P(a)$ for Scenarios 1 and 2 for the models for seroconversion with urogenital exposure in the supplementary material?**

[Response 4.9] *We thank the reviewer for noting this, we have now provided the analytical expressions on pages 1-2 of the supplementary information.*

[4.10] **Ditto for all 3 scenarios for the models for antibody acquisition with urogenital exposure? I encourage the authors to do so. In doing so, it would be much help if some non-trivial steps, where applicable, are also provided.**

[Response 4.10] *We thank the reviewer for raising this point. We have now provided the analytic expressions for the antibody acquisition models in the supplementary information, these can be found on pages 2-4.*

[4.11] **How an effective size of >350 was arrived at? Any justification for using less or more than this cutoff value?**

[Response 4.11] *On the basis that general ESS guidelines recommend >200 (http://beast.community/ess_tutorial) we initially aimed for at least 200, but increased it to 350, to ensure that we were not being too liberal – whilst ensuring that computational time wasn't wasted.*

[4.12] **Why the posterior distributions of the parameters of the best selected model for a study site is not shown in the supplementary material?**

[Response 4.12] *Priors for all model parameters were assumed to be uniformly distributed, we have now stated this clearly and provided the ranges for each of the distributions in the supplementary material, which can now be found on page 4.*

"Parameter estimation by MCMC was performed assuming uniform priors for each parameter. Ranges for each of the prior distributions were consistent across all models and study sites evaluated, and were as follows for the sero-prevalence models: $\lambda_t \sim U(0,10)$, $t_c \sim$

$U(0,90)$ (upper range was set to the maximum age of an individual within the dataset), $\rho \sim U(0,10)$, $\lambda_{ug} \sim U(0,10)$ and $\gamma \sim U(0,1)$. For the antibody acquisition model: $\alpha_t \sim U(0,1000)$, $\sigma \sim U(0,1)$, $r \sim U(0,10)$, $\alpha_{ug} \sim U(0,1000)$, $t_c \sim U(0,90)$ (upper range was set to the maximum age of an individual within the dataset).

For the estimate of each parameter we have provided the median and the 95% credible intervals which adequately describe the marginal posterior for each model fitted to each study site. Additionally, as we have already provided two assessments of chain mixing and convergence for each of the best fitting models we do not feel that additionally presenting the posterior distributions provides any value for the reader given the volume of information already presented in the supplementary information many density plots of posterior densities would be over-kill.

[4.13] **Lastly, I do not see the final take away message, clearly spelled out in the discussion section.**

[Response 4.13] We thank the reviewer for this helpful comment. A similar comment was provided by reviewer 2, and we have therefore re-structured the first paragraph of the discussion to clearly highlight the key findings from the manuscript. On page 8, lines 200-211 we now say the following:

“Serological surveillance for trachoma is being considered to help programmes undertake post-validation surveillance. Prior analyses of the potential utility of serology have used only single-study-site data, with few studies taking a quantitative modelling approach^{14,19,20}. In all settings evaluated, our modelling suggested transmission of trachoma had declined over time, either through a step-change in transmission, or more linearly. We estimated SRR half-life to be 26 (95% CrI: 21-34) years and 40 (95% CrI: 31-52) years, with an estimated half-life of the antibody response to be 7.3 (95% CrI: 6.5-8.2) years and 5.5 (95% CrI: 4.7-6.3) years, for Pgp3 and CT694 respectively: the first estimates published for these parameters. Our results suggested that SCRs below 0.015 year^{-1} correspond to TF <5%, and that the mean sero-prevalence for 1–9-year-olds when TF <5% was <7%. Whilst more data are needed to reduce the uncertainty in the relationship between sero-prevalence and TF prevalence, we present an initial operational threshold for interpreting serological data in trachoma programmes.”

[4.14] **In addition, the authors say "modelling suggested transmission of trachoma had declined over time" (at line 183 in the text). How one could verify this statement in the field? And which sites (with which intervention characteristics, such as, high MDA coverage, etc.) indicated better decline than the others?**

[Response 4.14] Coverage data for trachoma MDA is consistently reported to be above or equal to the targeted 80% level, therefore with the currently available data it would not be possible to make inference on what elements of the intervention were key to programmatic success without additional epidemiological information. We have however highlighted in the discussion how findings from serological data could be verified with other sources of epidemiological surveillance data. On pages 8-9, lines 224-231 we now say the following.

“Modelling suggested that transmission had declined in all regions. Validation of this could be undertaken by comparing these findings with longitudinal trachoma surveillance. We would expect to see estimates of declining transmission reflected by reductions in Ct PCR positivity and TF prevalence between the two cross-sections. In populations from which historical data (or multiple cross-sections) were available for this study (Nepal, The Gambia and Rombo) we see clear declines in TF and PCR prevalence (where available) mirroring the

serological pattern^{16,17,26}. For future studies, it will be important to ensure that long-term TF and PCR data are generated in order to validate sero-surveillance data."

[4.15] The last paragraph (lines 269 - 276) of the discussion section is very general. In the sentence "... here we have demonstrated the potential utility of antibody based surveillance to help monitor infection recrudescence after validation.", what does validation stand for? It is neither clear how one should design antibody data collection in the post-MDA phase of elimination programmes.

[Response 4.15] We have taken this comment on board made by the reviewer and rephrased our interpretation of the findings to highlight the idea that anti-body based surveillance could be helpful for monitoring low transmission levels. In the current phase of work, the purpose of research is primarily to see if antibody based surveillance can tell us anything about trachoma transmission in different endemicity regions, before we can advise on how data should be collected moving forwards. Sero-surveillance is currently not an established tool for trachoma programmes therefore it is important to understand how findings from the analysis of the data relate to the better established approach (repeated estimation of the prevalence of TF in children) already used for surveillance. Additionally, as the use of multiplex technology expands, in order to simultaneously monitor exposure to multiple pathogens, opportunities for this type of analysis is likely to increase. On pages 10-11, lines 298-310 we now say the following to clarify things:

"However, here we have demonstrated the potential utility of antibody-based surveillance to help monitor low-levels of ocular Ct infection transmission. We have initiated an evidence base for the use of sero-surveillance by programmes in low-transmission and post-elimination settings (where TF <5%), and have provided an operational threshold for sero-surveillance. Across a number of infectious diseases there is now a concerted effort towards integrated sero-surveillance, where samples from the same cross-sectional survey are tested for antibody responses to multiple pathogens simultaneously, using a multiplexed bead-based immunoassay platform³⁵. Given the enormous potential of integrated surveillance afforded by multiplexing, it is likely that an increasing amount of data of the format analysed here will become available in the future. This will increase the need for appropriate analytic methods. The approaches applied here could be adopted for other diseases, to help generate understanding of low-transmission scenarios, determine how such data can complement existing programme evaluation methods and guide future study design in the post-MDA phase."

[4.16] The authors say that at line 341 "All code is available to editors and referees upon request." My comment is that the computer codes used, along with any input files, should be made available to editors and reviewers without their request in order for facilitating a rigorous review process.

[Response 4.16] As per the reviewers request all data and code has been placed in into a publically accessible GitHub folder, these can be downloaded by anyone and are freely available. All files can be found here: https://github.com/Pinzo1/Serology_code-

Reviewers' Comments:

Reviewer #1:

None

Reviewer #3:

Remarks to the Author:

The authors have done a good job taking into account the various referees reports and carefully replying to each of the points made, also revising the MS accordingly. Therefore I have no further comments to made.

Reviewer #4:

Remarks to the Author:

I read the authors' rebuttal letter with interest and am happy to say that the authors have responded very well to the comments made by all the reviewers. As a result, the revision has definitely improved the manuscript in terms of a better presentation of the results and discussion points. However, I still think that the authors need to work on some of the comments raised by Reviewer 4.

1) The authors in Response [4.5] say that "... that changing from 1 to multiple axis labels is a comment pertaining primarily to style and taste." I agree with the authors. But increasing the size of the labels did not help the readability of Figure S1. I struggled to read the y-labels. I also asked my daughter, a 7th grader, to check if she could read them. I ask the authors to take a print-out of the PDF page in question and see whether they can read the labels.

2) I did not see the expressions for $P(a)$, although the authors said, in Response [4.9] and [4.10], that they had provided the analytical expressions.

3) Regarding Response [4.12]. The median and 95% credible intervals do not tell the whole story of the posterior distributions. I believe the authors must have looked at the plots of posterior densities of the best fitting parameters. Did any of the model parameters change significantly from the non-informative priors used? If they did, what significance they might have for trachoma epidemiology and elimination? I think the reviewer's comment was not meant for creating an over-kill in supplementary information. Rather it was to find out which of the model parameters changed significantly from their flat priors and potential implications for trachoma epidemiology.

Point by point response to the reviewers comments for manuscript NCOMMS-18-15959A

We thank the reviewers for the helpful and insightful comments they have provided on the manuscript. We have reproduced all reviewer comments in **bold** and our point by point response to each comment is provided in *italics*.

Reviewer #3 (Remarks to the Author):

[1.1] The authors have done a good job taking into account the various referees reports and carefully replying to each of the points made, also revising the MS accordingly. Therefore I have no further comments to made.

[Response 1.1] We thank the reviewer for their helpful comments are glad we have satisfied them.

Reviewer #4 (Remarks to the Author):

[2.1] I read the authors' rebuttal letter with interest and am happy to say that the authors have responded very well to the comments made by all the reviewers. As a result, the revision has definitely improved the manuscript in terms of a better presentation of the results and discussion points. However, I still think that the authors need to work on some of the comments raised by Reviewer 4.

[Response 2.1] We thank the reviewer for their helpful comments. We hope to address the final outstanding points in our response below.

[2.2] The authors in Response [4.5] say that "... that changing from 1 to multiple axis labels is a comment pertaining primarily to style and taste." I agree with the authors. But increasing the size of the labels did not help the readability of Figure S1. I struggled to read the y-labels. I also asked my daughter, a 7th grader, to check if she could read them. I ask the authors to take a print-out of the PDF page in question and see whether they can read the labels.

[Response 2.2] We have re-made Supplementary Figure 1 and we hope that the image is now clearer.

[2.3] I did not see the expressions for $P(a)$, although the authors said, in Response [4.9] and [4.10], that they had provided the analytical expressions.

[Response 2.3] Our apologies, these seem to have been misplaced in the re-submitted document. The expressions for $P(a)$ for the sero-catalytic and antibody acquisition models can be found on pages 28-33 of the Supplementary Methods.

[2.4] Regarding Response [4.12]. The median and 95% credible intervals do not tell the whole story of the posterior distributions. I believe the authors must have looked at the plots of posterior densities of the best fitting parameters. Did any of the model parameters change significantly from the non-informative priors used? If they did, what significance they might have for trachoma epidemiology and elimination? I think the reviewer's comment was not meant for creating an over-kill in supplementary information. Rather it was to find out which of the model parameters changed significantly from their flat priors and potential implications for trachoma epidemiology.

[Response 2.4] *The reviewer is correct that the median and 95% CrI do not tell the whole story of the posterior distributions, but they are effective and well recognised summary statistics. All parameters had uniform prior distributions* (essentially uninformative), i.e. no specific prior assumptions about the epidemiology of trachoma were assumed, beyond the structural constraint of the model variant under consideration. As part of the MCMC fitting procedure the estimated posteriors were systematically plotted against the assumed priors, and in all cases there were substantial differences between the priors and posteriors indicating that the data were informative. Including all of these diagnostic plots would certainly overwhelm the supplementary information.*